# Distinct H3K27me3 and H3K27ac Modifications in Neural Tube Defects Induced by Benzo[a]pyrene

**DOI:** 10.3390/brainsci13020334

**Published:** 2023-02-15

**Authors:** Shanshan Lin, Chengrui Wang, Zhiwen Li, Xiu Qiu

**Affiliations:** 1Division of Birth Cohort Study, Guangzhou Women and Children’s Medical Center, Guangzhou Medical University, Guangzhou 510623, China; 2Key Laboratory of Reproductive Health, Institute of Reproductive and Child Health, National Health Commission of the China, Beijing 100191, China; 3Department of Epidemiology and Biostatistics, School of Public Health, Peking University, Beijing 100191, China; 4Department of Women’s Health, Guangdong Provincial Key Clinical Specialty of Woman and Child Health, Guangzhou Women and Children’s Medical Center, Guangzhou Medical University, Guangzhou 510623, China; 5Provincial Clinical Research Center for Child Health, Guangzhou 510623, China; 6Guangzhou Women and Children’s Medical Center, Guangzhou Medical University, Guangzhou 510623, China; 7Disease and Department of Pediatric Surgery, Guangzhou Women and Children’s Medical Center, Guangzhou Medical University, Guangzhou 510623, China

**Keywords:** neural tube defects, CUT&Tag, RNA-seq, epigenetics

## Abstract

The pathological mechanisms of neural tube defects (NTDs) are not yet fully understood. Although the dysregulation of histone modification in NTDs is recognized, it remains to be fully elucidated on a genome-wide level. We profiled genome-wide H3K27me3 and H3K27ac occupancy by CUT&Tag in neural tissues from ICR mouse embryos with benzo[a]pyrene (BaP)-induced NTDs (250 mg kg^−1^) at E9.5. Furthermore, we performed RNA sequencing (RNA-seq) to investigate the regulation of histone modifications on gene expressions. Gene ontology and KEGG analysis were conducted to predict pathways involved in the development of NTDs. Our analysis of histone 3 lysine 27 modification in BaP-NTD neural tissues compared to BaP-nonNTD revealed 6045 differentially trimethylated regions and 3104 acetylated regions throughout the genome, respectively. The functional analysis identified a number of pathways uniquely enriched for BaP-NTD embryos, including known neurodevelopment related pathways such as anterior/posterior pattern specification, ephrin receptor signaling pathway, neuron migration and neuron differentiation. RNA-seq identified 423 differentially expressed genes (DEGs) between BaP-NTD and BaP-nonNTD group. The combination analysis of CUT&Tag and RNA-seq found that 55 DEGs were modified by H3K27me3 and 25 by H3K27ac in BaP-NTD, respectively. In the transcriptional regulatory network, transcriptional factors including Srsf1, Ume6, Zbtb7b, and Cad were predicated to be involved in gene expression regulation. In conclusion, our results provide an overview of histone modifications during neural tube closure and demonstrate a key role of genome-wide alterations in H3K27me3 and H3K27ac in NTDs corresponding with changes in transcription profiles.

## 1. Introduction

Neural tube defects (NTDs) are among the most common and serious birth defects, caused by the failure of the neural tube closure between 21 and 28 days after conception [1]. According to the World Health Organization, approximately 300,000 babies are born each year with NTDs [2], resulting in approximately 88,000 deaths and 8.6 million disability-adjusted life years [3,4]. In low-income countries, NTDs may account for 29% of neonatal deaths due to observable birth defects [5]. While folic acid supplementation has reduced the frequency of NTDs in clinical studies and myo-inositol can prevent some folate-resistant NTDs in mice, they remain a significant human health concern [6,7].

The etiology of NTDs is complex, involving genetic susceptibilities, environmental factors and unappreciated genetic-environmental interactions [1]. Over 200 genes have been found to cause or contribute to increased risk of NTDs in mouse or human [8]. Lower blood levels of the B-vitamin folate, exposure to valproic acid, maternal diabetes, maternal obesity and exposure to high temperatures during early pregnancy are well-recognized risk factors for NTDs [1,9]. Our previous studies showed that the polycyclic aromatic hydrocarbons (PAHs) contributed to the development of NTDs based on data from both humans and animals [10,11]. PAHs are emitted from the incomplete combustion of fossil fuels or biomass and are one of the main adverse organic chemicals in air pollution. Although increasing causes were reported for NTDs, the cellular and molecular basis of neural tube development remains not quite clear.

Recently, increasing lines of evidence have indicated the contribution of aberrant histone modifications to NTDs. Histone modifications, one of the most studied epigenetic modifications, play key roles in regulating gene expression and are relatively sensitive to environmental factors. Bai et al. reported decreased trimethylation of histone 3 lysine 27 (H3K27me3) in Wnt2b, Wnt7b and Igf2 in NTD affected embryos induced by retinoic acid [12,13]. Wan et al. observed reduced histone H3K27 acetylation in NTD mice in TNF-related genes, including Tnf and Casp8 [14]. Moreover, increased histone acetylation of Sirt2 and Sirt6 was suggested to be involved in diabetes-induced NTDs [15]. Evidence based on human fetal brain tissues showed that the bindings of dimethylation of histone 3 lysine 79 (H3K79me2) to Sufu, Rara and Itga3 were significantly altered in NTDs [16]. Zhang et al. has proven the role of homocysteinylation of H3K79 in human NTDs through the regulation of NTC-related genes including Cecr2, Smarca4, and Dnmt3b [17]. However, the whole picture of histone modifications during the stage of neural tube closure is scarce.

Therefore, in this study, our objective was to characterize the modifications to the genome-wide patterns of two key histone modifiers, H3K27me3 (marks associated with repressive regulatory regions) and acetylation of histone 3 lysine 27 (H3K27ac, marks associated with active regulatory regions) associated with NTDs. Specifically, we combined Cleavage Under Targets and Tagmentation (CUT&Tag), which can analyze histone modifications in a small number of cells [18], and RNA-sequencing (RNA-seq) to assess differences in epigenetic and transcriptional patterns in the development of NTDs with the use of our previous benzo[a]pyrene (BaP) (one of the common PAHs with high toxicity) induced-NTD mouse model.

## 2. Materials and Methods

### 2.1. Animal Experiments

The BaP-induced NTD mouse model was performed as previously described [10]. Briefly, ICR mice of 8–9 weeks old weighing 28 ± 2 g were used in the experiment. Female mice were mated with males overnight and vaginal plugs were examined the following morning. Noon on the day of finding a vaginal plug was considered 0.5 days of embryonic development (E0.5). Pregnant mice were randomly divided into 2 groups. In BaP-treatment groups, mice were given BaP (CAS-No. 192-97-2, purity 98%, Sigma-Aldrich, St. Louis, MO, USA) intraperitoneally, dissolved in corn oil (CAS-No. 8001-30-7, aladdin, Shanghai, China) (25 mg mL^−1^), from E6.5 for three consecutive days at a dose of 250 mg kg^−1^. Mice in the control group were treated with corn oil (10 mL kg^−1^). On E9.5, pregnant mice were sacrificed by cervical dislocation and the embryos were removed by caesarean section. Embryos were carefully inspected for visible external malformations under a dissecting microscope. NTD-affected embryos were classified as showing distinct evidence of failed closure of the cephalic neural tube. Finally, 16.4% of the embryos were observed with open neural tube (Appendix A). The somite numbers were carefully counted. E9.5 embryos with 22–27 somites were used for further analysis [19].

### 2.2. Isolation of Neural Tissues

In BaP-treatment group, E9.5 embryos affected with cephalic NTDs were collected as BaP-NTD group, and embryos from the same dam without obvious malformations were collected as BaP-nonNTD group. Embryos with no malformations from the control group were collected as control group. Specially, neural tissues of five E9.5 embryos collected from five independent dams were pooled as one sample for RNA-seq experiment, and neural tissues from fourteen embryos collected from seven independent dams were pooled as one sample for CUT&Tag experiment. Embryonic neural tissues, from the most rostral aspect of the forebrain to the caudal aspect of the hindbrain, were excised according to the previous study [20] and further trimmed to eliminate any non-neural tissues as precisely as possible, the precise scope was shown in Figure 1. Three biological replicates for each group were conducted for RNA-seq and CUT&Tag, respectively.

### 2.3. RNA-seq

Total RNA was extracted from neural tissues using the TRIZOL method and the concentration was determined using NanoDrop 2000 (Thermo Fisher Scientific, Waltham, MA, USA). Library preparation was performed using the Illumina’s NEBNext^®^ UltraTM RNA Library Prep Kit (Illumina, San Diego, CA, USA) and high-throughput RNA-seq was performed using the Illumina Novaseq 6000 platform (Illumina, San Diego, CA, USA) with 100 bp paired-end sequencing.

### 2.4. Analysis of Differential Gene Expression and Functional Annotation

RNA-seq raw reads were removed adapters and trimmed low quality reads with Trimmomatic v.0.36. These processed reads were then mapped to the mouse genome (GRCm38/mm 10) using Hisat2 software. The readings were counted using HTseq-count and normalized to fragments per kilobase (FPKM) for further visualization [21]. The differentially expressed genes (DEGs) were identified using the DESeq2 package and defined as a fold change (FC) >2 and a *p*-value < 0.05. The DEGs were then subjected to Gene Ontology (GO) and pathway enrichment analysis to identify the significantly enriched functional classification, signaling and metabolic pathways, which were performed based on Gene Ontology Database (http://www.geneontology.org/, accessed on 1 January 2019) [22] and Kyoto Encyclopedia of Genes and Genomes (KEGG) pathway database (http://www.genome.jp/kegg/, accessed on 1 June 2022) [23] with R package clusterProfiler. GO terms were identified to be significantly enriched when *p* < 0.05 after Benjamini multiple test correction. Pathways were identified to be significantly enriched when false discovery rate (FDR) < 0.05.

### 2.5. CUT&Tag Experiment

We performed chromatin profiling with CUT&Tag according to its standard protocol (https://www.protocols.io/view/bench-top-cut-amp-tag-kqdg34qdpl25/v3?version_warning=no&step=69, accessed on 26 February 2020) [18] with NovaNGS CUT&Tag 3.0 High-Sensitivity Kit (for Illumina) (Novoprotein, Shanghai, China). In brief, neural tissues were homogenized and followed by incubation with Concanavalin A-coated Magnetic Beads (Novoprotein, Shanghai, China) for 10 min at room temperature. Next, H3K27me3 rabbit pAb (1:100, Sigma-Aldrich, St. Louis, MO, USA) and H3K27ac rabbit pAb (1:100, Abcam, Cambridge, MA, USA) were employed as primary antibodies, respectively, and Goat Anti-Rabbit IgG H&L (1:100, Abcam, Cambridge, MA, USA) was used to bind primary antibodies. After binding pG-Tn5 adapter complex (Novoprotein, Shanghai, China), tagmentation was conducted. Following library amplification, DNA quantity was determined with Qubit 4 (Thermo Fisher Scientific, Waltham, MA, USA) and library quality characterized with Agilent TapeStation 4200. All libraries were sequenced using NovaSeq 6000 (Illumina, San Diego, CA, USA).

### 2.6. CUT&Tag Data Processing and Analysis

We used FastQC to check the quality of the raw reads. Low quality reads as more than 10% of bases with quality scores less than 20 were filtered. The remaining reads that passed all the filtering steps were counted as clean reads and aligned to the mouse genome (GRCm38/mm 10 build) with the BWA mem v.0.7.12. Only non-redundant and uniquely mapped reads were retained to correct for sequencing bias. Peaks and enriched regions were called with MACS2 v.2.1.0 [24]. Noisy peaks with very weak signals were removed in further analysis. PeakAnnotator was used to identify the nearest TSS of every peak and the distance distribution between peaks and TSS was shown. Moreover, the distribution of peak summits on different function regions, such as 5utr, CDS, 3utr, was performed. Different peak analysis was based on the software PePr v.1.1.18 and defined as the FC between two groups being more than 2, *p* < 1 × 10^−5^ [24]. Peak-related genes were confirmed by PeakAnnotator. Data were visualized in the UCSC genome browser. Functional analyses were performed using the edgeR package. Hypergeometric optimization of motif enrichment (HOMER) was used to discover motifs of specific regions [25].

### 2.7. Quantitative Real-Time Quantitative Polymerase Chain Reaction (qRT-PCR)

RNA was isolated from neural tissues of E9.5 embryos using Trizol (Invitrogen, Carlsbad, CA, USA); genomic DNA was removed by DNase I digestion (DNA-free, Invitrogen, Carlsbad, CA, USA) and then reverse-transcribed using random hexamers (Superscript VILO cDNA synthesis kit, Invitrogen, Carlsbad, CA, USA). The abundance of mRNA of Myl2, Csrp3 and Grm2 were analysed using real-time PCR (iTaqTM Universal SYBR Green Supermix, BioRad, Hercules, CA, USA) on a 7500 Fast Real Time PCR system (Applied Biosystems, Thermo Fisher Scientific, Waltham, MA, USA), with each sample analysed in triplicate. Primers are listed in Appendix A. Relative quantification of each gene expression level was normalized according to the Gapdh gene expression.

### 2.8. Statistical Analyses

Statistical analyses were carried out using SPSS version 20.0 (IBM, Chicago, IL, USA). Non-paired Student’s *t*-test was used when two groups were being compared. Differences were considered statistically significant at *p* < 0.05. Data are represented as the mean ± standard error of mean (SEM) of at least in triplicate.

## 3. Results

### 3.1. CUT&Tag Analysis

Neural tissues from three experimental groups: NTDs in BaP exposed group (BaP-NTD), non-NTDs in BaP exposed group (BaP-nonNTD), and normal embryos in control group (control) were extracted to profile epigenetic changes (Figure 1a). The quality of the clean reads and mapping rates of CUT&Tag data are shown in Appendix A. Figure 1b-d summarized differences in H3K27me3 and H3K27ac abundance between the three different experimental groups. BaP exposure significantly decreased the total number of H3K27me3 peaks, while had little impact on the number of H3K27ac peaks. The H3K27me3 and H3K27ac peak widths in BaP-NTD group were wider than those in control and BaP-nonNTD group (Appendix A). The distribution of genomic regions modified by H3K27me3 and H3K27ac was classified into eight regions (promoters, upstream2k, 5utr, exons, introns, 3utr, downstream2k and cgi) according to the UCSC Genome Browser. The H3K27me3 and H3K27ac marks were mainly mapped to intron and promoter regions (Figure 1d). BaP exposure resulted in decreased H3K27me3 peaks in upstream2k, intron and cig, while it had little to no impact on H3K27ac peaks across the whole gene (Appendix A). No significant differences on H3K27me3 or H3K27ac peak distributions were observed between BaP-NTD and BaP-nonNTD embryos.

As shown in Figure 2a,b, there were unique H3K27me3 binding genes present in each group. Compared to controls, 397 mutual H3K27me3 and 226 H3K27ac binding genes were identified in both BaP-NTD and BaP-nonNTD group, and 895 H3K27me3 and 431 H3K27ac binding genes were unique for BaP-NTD group. The GO enrichment analysis showed that these 397 mutual H3K27me3 peak related genes were mainly involved in synapse organization, axonogensis, dendrite development and postsynaptic membrane (Figure 2c). Similar enrichment terms were observed for the 226 mutual H3K27ac binding genes (Figure 2d, Appendix A). Function annotation of the 895 H3K27me3 binding genes uniquely associated with BaP-NTD group revealed a significant enrichment of terms associated with neurodevelopment including growth cone, site of polarized growth and distal axon (Figure 2e, Appendix A). However, no significant enrichment pathways were observed for these 431 H3K27ac unique binding genes (Figure 2f, Appendix A). The top 10 unique peak binding genes identified in BaP-NTD are listed in Figure 2g,h.

A total of 6879 and 9914 differential H3K27me3 peaks, and 1574 and 3753 differential H3K27ac peaks were identified in BaP-NTD/control and BaP-nonNTD/control comparison, respectively (Appendix A). The overview of the enriched pathways based on these differential peak-associated genes is presented in Figure 3a. Compared to BaP-nonNTD/control comparison, more pathways associated with neural development were enriched for BaP-NTD/control comparison. When compared the histone modification between BaP-NTD and BaP-nonNTD, 6045 differential H3K27me3 peaks and 3104 differential H3K27ac peaks were identified (Appendix A). As shown in Figure 3c,d, nerve system development was enriched as one of the top terms for both the differential H3K27me3 and H3K27ac binding genes for BaP-NTD/BaP-nonNTD comparison. Other classical neurodevelopment related pathways enriched included neurogenesis, neuron migration, neuron differentiation, neural crest cell migration, axon guidance, axonogenesis, brain development, brain morphogenesis, forebrain development, cerebellum development, cerebral cortex development, anterior/posterior pattern specification, ephrin receptor signaling pathway (Appendix A). Representative differential H3K27me3 and H3K27ac peaks between BaP-NTD and BaP-nonNTD are shown in Figure 3b.

### 3.2. Transcriptome Analysis

To further investigate how H3K27 modification status links with individual gene expression, we created a total RNA-seq data. An overview of the RNA-seq data is presented in Appendix A. The relationship between histone modifications and gene expressions across all the three groups is presented in Figure 4a. As expected, genes with an H3K27me3 mark in the promoter region had a lower expression, those with an H3K27ac mark had a higher expression, and those with both marks had a distribution of expression levels somewhere in between.

A total of 1501 DEGs were observed for BaP-NTD/control comparison, which was significantly higher than that for BaP-nonNTD/control comparison (673 DEGs) (Figure 4b and Appendix A). The expressions of five representative neural tube closure related genes among the three groups are shown in Figure 4c. Compared to the control group, the expressions of Frem2, Vangl1, Ptk7, were significantly increased, while the expressions of Pax7, and Fkbp8 were decreased in the BaP-NTD group. In contrast, the levels of these genes in the BaP-nonNTD group were somewhere in between, and 423 DEGs were found for BaP-NTD/BaP-nonNTD comparison (Figure 4b, Appendix A). Overall, biological function analysis revealed that these 423 DEGs were mainly classified as three classes: (1) genes involved in the cellular process/metabolic process, mainly for neurodevelopment (314 genes), (2) genes involved in multicellular organismal process (206 genes), and one of the top terms was nervous system development (3) genes involved in localization (153 genes) (Figure 4d,e).

### 3.3. Combined Analysis of Histone Modifications and Transcription

If we further characterize the DEGs with differential H3K27 modifications, no apparent trends were observed between gene expressions with histone modifications (Figure 5a, Appendix A). A total of 198 DEGs with H3K27me3 modification and 65 DEGs with H3K27ac modification unique for BaP-NTD/control comparison were observed (Figure 5b, Appendix A). The expressions of three top genes, Myl2, Csrp3 and Grm2, were validated by qRT-PCR (Appendix A). When compared BaP-NTD to BaP-nonNTD, 55 DEGs with H3K27me3 modification and 25 DEGs with H3K27ac modification were observed (Appendix A). We then performed GO analysis to investigate the biological function and discovered that these 55 DEGs with peaks of H3K27me3 were highly enriched in growth and developmental growth, and these 25 DEGs with H3K27ac modification were enriched in growth and neuron differentiation (Figure 5c). Combining the 263 DEGs unique for BaP-NTD/control comparison with the 80 DEGs for BaP-NTD/BaP-nonNTD comparison, 19 common DEGs including Myl2, Dlg2 and Rgs6 were observed (Table 1).

### 3.4. Motif Analysis

We further performed motif analysis to identify the potential transcription factors which may regulate the DEGs in BaP-NTDs by HOMER, the de novo motif analysis. As shown in Figure 5d, five motifs were significantly (threshold, *p*-value < 1 × 10^−12^) enriched in upregulated genes associated with increased H3K27me3 peaks, and two motifs in upregulated genes with decreased H3K27me3 peaks for BaP-NTD/control comparison. Of these, Smad2 and SRSF1 were previously shown to be involved in the development of NTDs. No significant transcription factor motifs were identified in other groups (Appendix A).

## 4. Discussion

In this study, we correlated histone modifications to gene expressions in mouse NTDs via integrative analysis of the RNA-seq and CUT&Tag data, with both internal controls and vehicle controls to provide new and comprehensive information on epigenetic regulation during neural tube closure. Our data indicated several genes with specific alterations of histone modifications in NTDs that might be responsible for the failure of neural tube closure, such as Myl2, Dlg2, and Rgs6. Function analysis revealed potential key pathways implicated in the pathological process of NTDs, such as anterior/posterior pattern specification, ephrin receptor signaling pathway, neuron migration and neuron differentiation. In addition, our analysis identified new transcription factors which may be involved in the epigenetic regulation of molecular events in BaP-induced NTDs, including Srsf1, Ume6, Zbtb7b, and Cad.

Previous findings suggested that PAHs exposure was associated with the occurrence of NTDs [11,26]. However, the underlying pathological mechanisms have not been fully elucidated. Neurulation is a complex and multistep process, involving the precise temporal and spatial regulation of gene expression [27]. A considerable number of biological events are involved in the process of neural tube closure, such as apoptosis, proliferation, differentiation and migration [27,28]. Thus, it is of great significance to explore the key molecular events and regulatory mechanisms that lead to the failure of neural tube closure. With the comparison of internal and vehicle control group, our results identified a number of pathways specially enriched in NTDs, such as anterior/posterior pattern specification, which was closely related with neurodevelopment [29]. Moreover, we found that the ephrin receptor signaling pathway might be implicated in BaP induced NTD. Ephrin-A5 is a recognized gene implicated in cranial NTD based on mice models [30]. Evidence was further presented that Ephrin/Eph-mediated signaling was tightly associated with neural tube anterior-posterior patterning [31]. However, this research was performed with vertebrate spinal cord. Whether Ephrin participated in anterior-posterior patterning during cranial neural tube closure required further studies.

In addition, the GO analysis in the present study revealed that compared to its internal controls, a number of pathways participated in neuron differentiation and migration were mainly enriched in embryos affected by NTDs, especially in the forebrain. Our previous work has shown that the majority of NTDs induced by BaP exposure occurred in the forebrain [10]. During embryonic development, the tightly regulated cellular events including proliferation, apoptosis, differentiation, and migration are crucial for neural tube development [32,33]. The involvement of neuron apoptosis in BaP-induced NTD was confirmed in our previous study [10]. Data presented here indicated that the disruption of neuron differentiation and migration might be also one of the main underlying mechanisms for neural tube closure failure caused by BaP. Recent studies have confirmed the effect of histone medication on neurulation through regulation of cell differentiation. One study from Patterson et al. found that in Mox2-Cre mutant embryos, proliferation and apoptosis was not affected in neural tissues, while the disruption of interneuron differentiation and specification was the major contributor to the observed NTDs, and they further suggested the involvement of histone acetylation in the regulation of neuronal differentiation [34]. Li et al. reported that the EZH2-mediated H3K27 methylation was responsible for the premature differentiation of neuron cells in zebrafish with deficits in forebrain caused by knockdown of Sox19b [35]. Another NTD zebrafish model induced by knockdown of Pcgf1 also indicated the role of histone modification in regulation of neuron differentiation [36]. However, relevant studies on the epigenetic regulation of neuron migration through histone modification in NTD were scarce. Further research is needed to fully clarify the underlying mechanisms of dysregulation of neuron differentiation and migration caused by histone modification in NTD.

Deciphering the molecular underpinnings of neural tube development would be helpful for providing new strategies for prevention. We identified a set of genes with changes of both transcriptions and histone modifications specifically in BaP-NTD group, some of which were reported to be implicated in neurological disorders. For example, Ctnna3 was implicated in Alzheimer’s disease [37]. Inhibition of Dpp4 showed a neuroprotection effect in diabetic mouse brain [38], and the Cacna2d2 mutant mice were affected with cerebellar neuro-degeneration associated with ataxia and seizures [39]. However, the role of these genes in neurodevelopment during embryonic stage was not clear. Moreover, we also found several candidate genes that were associated with neuron development. For example, our data showed an increased expression of Dlg2 with decreased of H3K27me3 specifically in BaP-NTD group, which was a known gene vital for neuronal differentiation, maturation, and migration [40]. Myl2 was reported as one of the main down regulated genes in In Adnp^−/−^ NTD mice, and Rgs6 was one of the upstream modulators of Notch signaling, and the mutant of which would result malformations including delayed neural tube maturation [41]. The present work indicated higher expression of Myl2 and Rgs6 with histone modification in BaP-NTD group compared to BaP-NTD and control group. Thus, our results implied that the up-regulation of these two genes might also cause NTD. Altogether, our work provided new evidence for the regulation role of H3K27 histone modifications in the etiology of NTDs and suggested potential new gene targets for NTDs pathogenesis.

Interestingly, by de novo motif analysis, we identified enrichment of sequence motifs unique for the differentially trimethylated regions in NTD embryos. Among them, Smad2 has been well demonstrated to be vital for neural tube closure by regulating neural epithelial organization, including tissue shape, size and three-dimensional morphogenesis through tight junction pathways [42]. In frog, Smad10 was required for the formation of nervous system, as a key component in the development of both anterior and posterior neural tissue [43]. Srsf1 has emerged as a key oncodriver in tumor progression such as gliomagenesis for its role in alternative splicing [44]. The target genes of Srsf1 included Stat3, Bcl-x, Mdm2 [45,46,47], which were all related to neural tube closure [48,49,50]. Moreover, Ume6 was used to be regarded as a key transcriptional regulator of early meiotic gene expression [51,52], and its role in autophagy has been documented recently [53]. Zbtb7b belongs to the large ZBTB family, known as critical factors regulating the lineage commitment and differentiation of hematopoietic-derived cells [54]. Cad is a multifunctional protein that participates in the initial three speed-limiting steps of pyrimidine nucleotide synthesis. The important role of Cad in neurological disorders including epileptic encephalopathy and Alzheimer’s disease has also been emphasized [55]. However, the direct molecular function of Ume6, Zbtb7b and Cad in neural tube closure has not been well characterized.

Few studies have examined the effects of BaP exposure on neurodevelopment in offspring without obvious NTDs. Air pollution continues to be a worldwide environmental health problem, which has an enormous influence on fetal development beginning during gestation. PAHs are one of the main toxic, mutagenic, and carcinogenic air pollutants and can cross the placenta. Our study found that although some embryos showed no obvious neural tube malformations after BaP treatment, their neurodevelopment were impaired as shown by the differentially expressed and histone modified neurodevelopment-related genes, such as wnt7b, Fzd and Cdh9. Epidemiological studies have demonstrated that PAHs exposure were positively correlated with the increased risks for neurobehavioral development problems, including attention deficit hyperactivity disorder [56], decrease of IQ [57], and cognitive developmental delay [58]. Animal experiments implied that the potential underlying mechanisms included oxidative stress, inflammation, altered levels of dopamine and/or glutamate, and changes in synaptic plasticity/structure [59]. The synapse is fundamental for the sophisticated ensemble of the highly specialized neuron network. The non-malformed BaP-treated embryos in the present study showed significant disturbances in pathways associated with synapse, including synapse organization, synapse assembly, chemical synaptic transmission and regulation of synaptic plasticity. Our data supported the previous results, emphasizing the role of synapse in PAHs-induced embryo neurotoxicity, and provided new insights for further study.

It should be mentioned that there are some limitations in this study. The candidate genes and related pathways found in our study were not further investigated. Functional analyses of key genes in further studies were required to obtain more in-depth understanding for development of NTDs. Another drawback is that we used intraperitoneal injection of BaP to induce NTDs in mice, while humans are mostly exposed to PAHs through inhalation and diet in the real world. Previous works from ours and others have tried to induce NTDs with BaP through gavage, however, few NTDs were observed [60,61]. We acknowledged that the pathogenesis of NTDs in mice induced by intraperitoneal injection of BaP could be different from that of human NTDs. However, many of the identified pathways and genes in the present study were reported to be implicated in other NTD animal models and human studies. Our previous studies have identified some key pathogenic genes and pathway shared between this BaP-NTD mouse model and human cases [10,62].

## 5. Conclusions

With this study, we showed that genome-wide H3K27me3 and H3K27ac modification profiles were useful to discover novel molecular pathways and genes for the better understanding of NTD pathophysiology. We would recommend future studies to confirm the newly identified targets in the present study. Moreover, since neural tissue samples represent an average state across different cellular compartments, newer techniques looking at subpopulations of cells might help to unravel cell-specific deregulated pathway.

## Figures and Tables

**Figure 1 brainsci-13-00334-f001:**
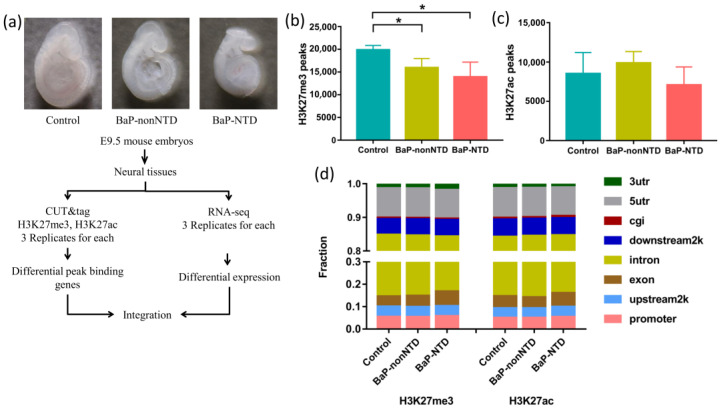
Histone modification profiling of H3K27me3 and H3K27ac in neural tissues from E9.5 normal mouse embryos and NTD embryos. (**a**) Schematic illustration of experimental design. The number of peaks called in different groups for H3K27me3 (**b**) and H3K27ac (**c**). (**d**) The genomic distribution of different histone modification regions in control, BaP-nonNTD and BaP-NTD embryos. * *p* < 0.05, compared with control group.

**Figure 2 brainsci-13-00334-f002:**
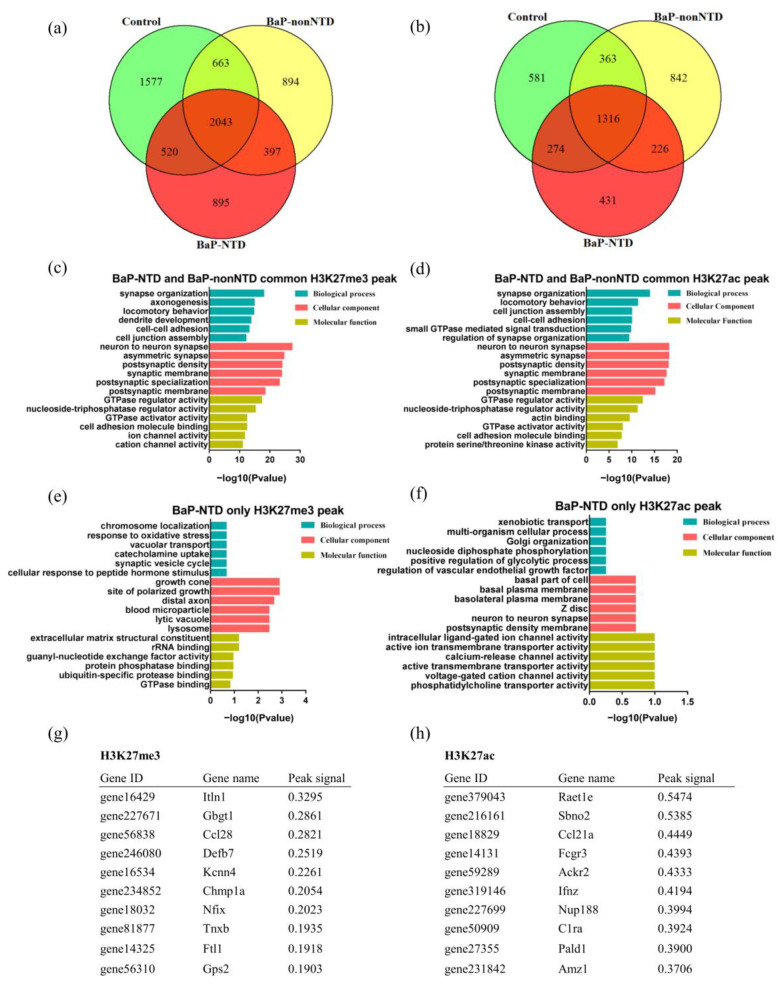
Common and unique H3K27me3 and H3K27ac peaks in E9.5 mouse embryos with and without NTDs. Venn diagram showing the overlap of H3K27me3 (**a**) and H3K27ac (**b**) binding genes. Top six GO biological processes, cellular components, and molecular functions enriched from common H3K27me3 (**c**) and H3K27ac (**d**) binding genes between BaP-nonNTD and BaP-NTD (ordered by significance level). Top six GO biological processes, cellular components, and molecular functions enriched from unique H3K27me3 (**e**) and H3K27ac (**f**) binding genes for BaP-NTD (ordered by significance level). The list of the top 10 unique BaP-NTD peak binding genes for H3K27me3 (**g**) and H3K27ac (**h**).

**Figure 3 brainsci-13-00334-f003:**
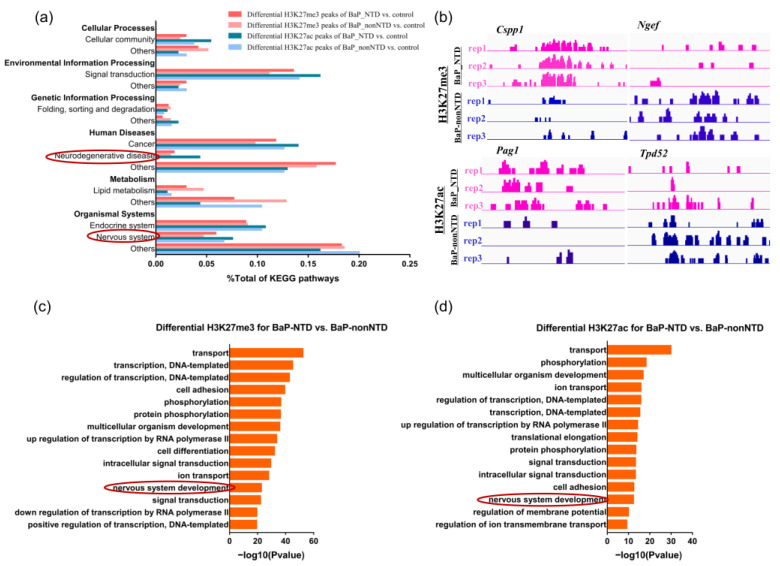
Analysis of differentially modified histone peaks in E9.5 mouse embryos with and without NTDs. (**a**) Overview of KEGG enrichment analysis of differentially modified histone peak binding genes identified in BaP-NTD-vs-control and BaP-nonNTD-vs-control comparisons. (**b**) Integrative genomics viewer snapshot of representative differential modified H3K27me3 and H3K27ac signals in comparison of BaP-NTD with BaP-nonNTD. Biological process GO enrichment terms for the differential H3K27me3 (**c**) and H3K27ac (**d**) peak binding genes identified in BaP-NTD-vs-BaP-nonNTD comparisons, ordered by adjusted *p*-value. Neurodevelopment associated terms were marked with red circles.

**Figure 4 brainsci-13-00334-f004:**
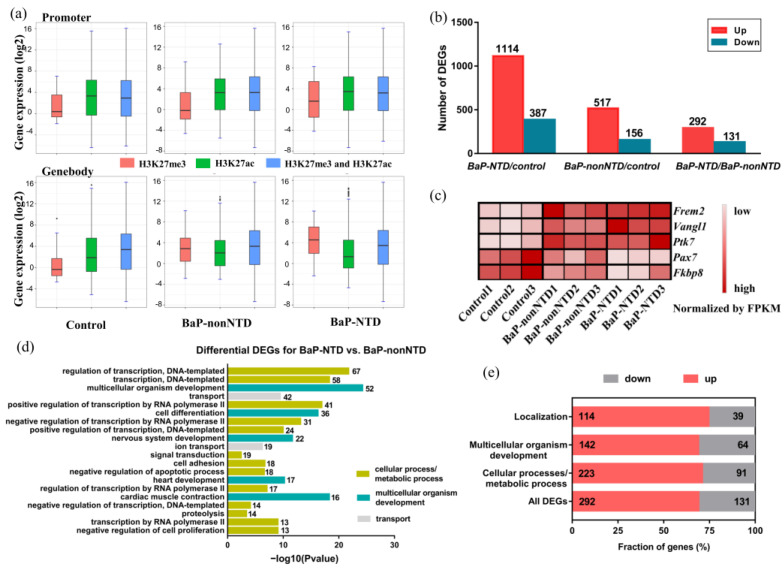
Transcriptome profile of E9.5 mouse embryos with and without NTD. (**a**) The relationships between histone modifications of H3K27me3/H3K27ac and gene expressions. (**b**) Number of differentially expressed genes identified in BaP-NTD-vs-control, BaP-nonNTD-vs-control, and BaP-NTD-vs-BaP-nonNTD comparisons. Red in each column represents the number of up-regulated genes. Green in each column represents the number of down-regulated genes. (**c**) Expression levels of five representative neural tube closure related genes in different groups. (**d**) Gene ontology analysis of common differentially expressed genes identified in BaP-NTD-vs- BaP-nonNTD comparisons. GO terms associated with cellular process/metabolic process (yellow), multicellular organismal process (blue), and localization (grey) were highlighted respectively. (**e**) Genes contributing to the three main GO term classes were divided into up-(red) and down-regulated (gray) fractions. Gene numbers in each fraction are indicated.

**Figure 5 brainsci-13-00334-f005:**
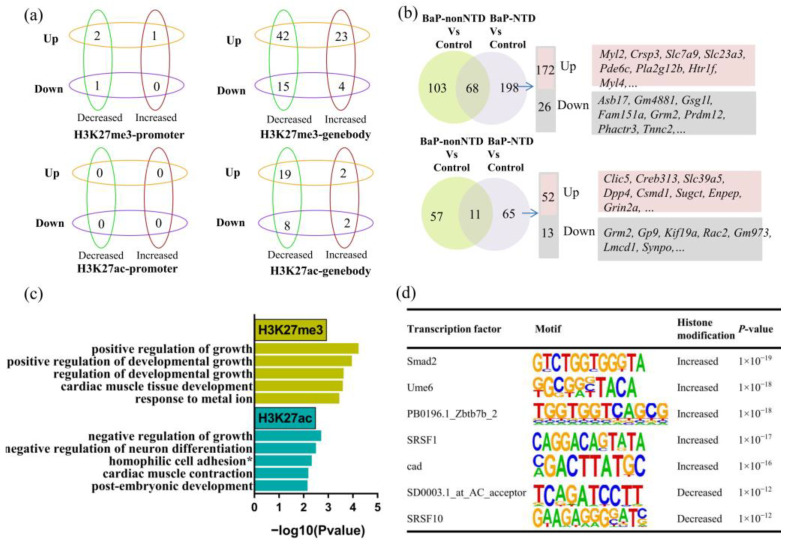
Combined analysis of CUT&Tag and RNA-seq. (**a**) The relationships between fold change of CUT&Tag signals for H3K27me3/H3K27ac and fold change of gene expression in BaP-NTD-vs-BaP-nonNTD comparison. (**b**) Venn diagram showing the overlap differentially transcribed genes with a H3K27me3 or H3K27ac modification. The BaP-NTD-vs-control comparisons unique DEGs are listed, ordered by fold change of gene expression. (**c**) GO analysis of the BaP-NTD-vs-BaP-nonNTD comparisons with a H3K27me3 or H3K27ac modification. (**d**) Analysis using HOMER for differentially modified H3K27me3 regions revealed eight significant motifs. ^*^ homophilic cell adhesion via plasma membrane adhesion molecules.

**Table 1 brainsci-13-00334-t001:** The differentially expressed genes with histone modifications unique for E9.5 BaP-NTD embryos.

Gene ID	Gene Name	Expression FC	Expression State	Histone Modification	Modification State	Location
17906	Myl2	6.787	Up-regulated	H3K27ac	Decreased	Genebody
383348	Kctd16	2.997	Up-regulated	H3K27ac	Increased	Genebody
65256	Asb2	2.159	Up-regulated	H3K27me3	Increased	Genebody
213435	Mylk3	2.084	Up-regulated	H3K27me3	Decreased	Genebody
19415	Rasal1	2.016	Up-regulated	H3K27me3	Increased	Genebody
13482	Dpp4	1.603	Up-regulated	H3K27me3	Decreased	Promoter
270049	Galntl6	1.484	Up-regulated	H3K27me3	Decreased	Genebody
23859	Dlg2	1.47	Up-regulated	H3K27me3	Decreased	Genebody
14799	Gria1	1.47	Up-regulated	H3K27me3	Increased	Genebody
74189	Phactr3	1.454	Down-regulated	H3K27me3	Decreased	Genebody
216033	Ctnna3	1.428	Up-regulated	H3K27me3	Increased	Genebody
545428	Ccdc141	1.269	Up-regulated	H3K27ac	Decreased	Genebody
56808	Cacna2d2	1.247	Up-regulated	H3K27me3	Decreased	Genebody
50779	Rgs6	1.246	Up-regulated	H3K27me3	Increased	Genebody
50787	Hs6st3	1.154	Up-regulated	H3K27ac	Decreased	Genebody
237558	Myrfl	1.088	Up-regulated	H3K27me3	Decreased	Genebody
22041	Trf	1.083	Up-regulated	H3K27me3	Increased	Genebody

FC, fold change.

## Data Availability

All data generated in this manuscript have been deposited in NCBI Sequence Read Archive under the accession number PRJNA875213 (https://dataview.ncbi.nlm.nih.gov/object/PRJNA875213?reviewer=q147jtjlbev36sl1htfi01nnff, accessed on 10 February 2023), PRJNA874853 (https://dataview.ncbi.nlm.nih.gov/object/PRJNA874853?reviewer=pl5qimnb04947llhcvi2peu7bs, accessed on 10 February 2023), and PRJNA874873 (https://dataview.ncbi.nlm.nih.gov/object/PRJNA874873?reviewer=o0oo847r8f8kgp77kgeo62c27p, accessed on 10 February 2023).

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
