# Peer review of "Distinct H3K27me3 and H3K27ac Modifications in Neural Tube Defects Induced by Benzo[a]pyrene"

_brainsci, 2023, doi:10.3390/brainsci13020334_

Round 1

Reviewer 1 Report

This manuscript tried to reveal the epigenetic dysregulation and following transcriptional changes in neural tissues using benzo(a) pyrene-induced NTD models. This study comprehensively analyzed methylated and acetylated histone modification and the differential gene expression on BaP-induced NTDs. However, the data was somehow biased due to a lack of consistency of control in dataset analysis. The authors should provide the comparison data analysis for BaP NTD over BaP non-NTD rather than BaP NTD over control for all the data set.  

Major concerns

1. Figure 1b: There is a significant difference in H3K27me3 in the BaP-treated group compared to the control group; however, it is unclear if there is any significant difference in H3K27me3 between BaP-non NTD group and BaP NTD group. Then what caused the phenotypic difference in BaP treated group? Does it mean that H3K27me3 and H3K27ac modifications are not critical for BaP-induced NTD phenotypes?  

2. Figure 3b: The comparison between BaP non-NTD and BaP NTD should be included. It should be more critical data that compares BaP non-NTD and BaP NTD over control.

3. BaP-induced NTDs presented with mainly cranial NTDs (anencephaly or exencephaly) rather than spinal NTDs. Still, your GO and pathway analysis is enriched in the spinal dorsoventral patterning, and the genes that the authors think top candidate genes, such as Fkbp8 or Vangl2, are related to spinal NTDs, rather than cranial NTDs. The authors should explain the discrepancy between phenotypes and the result of the data set analysis.

Minor:

The labeling in Figures is hard to read. Please improve the resolution of images or increase the font size in the Figures.

Reviewer 2 Report

This is a very interesting manuscript that uses multi-omic approaches to profile the genome-wide H3K27me3 and H3K27ac occupancy and changes in gene expression in mouse neural tissues with BaP-induced NTDs.  They identify 100’s of differentially expressed genes that are modified by H3K27me3 and 67 by H3K27ac in BaP-NTD. These include multiple genes implicated in NTDs in animal models.   The manuscript is well-written, and the data is well-presented. I only have 2 issues to raise. 

The title implies that the paper will bring mechanistic insight to NTDs.  However, the data presented is only correlative identifying differences that occur in BaP-NTDs, and no mechanistic studies are performed to demonstrate that the differences in these pathways identified actually lead to NTDs with BaP exposure. Thus “mechanistic” in the title is overstated.   

In the discussion, when describing the function of the various differentially expressed genes in BaP-NTDs, processes that occur after neural tube closure.  For instance, neuron projection occurs after neural tube closure and cannot be involved. Similarly, synapse organization, synapse assembly, chemical synaptic transmission, and regulation 435 of synaptic plasticity occur after the neural tube has closed.

Round 2

Reviewer 1 Report

Thank you for your extensive revision.

The authors addressed all raised issues.